# Enhanced treatment strategies and distinct disease outcomes among autoantibody-positive and -negative rheumatoid arthritis patients over 25 years: A longitudinal cohort study in the Netherlands

**Xanthe M. E. Matthijssen** *, **Ellis Niemantsverdriet** , **Tom W. J. Huizinga, Annette H. M. van der Helm–van Mil**

Department of Rheumatology, Leiden University Medical Center, Leiden, Netherlands

* x.m.e.matthijssen@lumc.nl

## Abstract

### Background

Based on different genetic and environmental risk factors and histology, it has been proposed that rheumatoid arthritis (RA) consists of 2 types: autoantibody-positive and autoantibody-negative RA. However, until now, this remained hypothetical. To assess this hypothesis, we studied whether the long-term outcomes differed for these 2 groups of RA patients.

### Methods and findings

In the Leiden Early Arthritis Clinic cohort, 1,285 consecutive RA patients were included between 1993 and 2016 and followed yearly. Treatment protocols in routine care improved over time, irrespective of autoantibody status, and 5 inclusion periods were used as instrumental variables: 1993–1996, delayed mild disease-modifying antirheumatic drug (DMARD) initiation (reference period); 1997–2000, early mild DMARDs; 2001–2005, early methotrexate; 2006–2010, early methotrexate followed by treat-to-target adjustments; 2011–2016, similar to 2006–2010 plus additional efforts for very early referral. Three long-term outcomes were studied: sustained DMARD-free remission (SDFR) (persistent absence of clinical synovitis after DMARD cessation), mortality, and functional disability measured by yearly Health Assessment Questionnaire (HAQ). Treatment response in the short term (disease activity) was measured by Disease Activity Score–28 with erythrocyte sedimentation rate (DAS28-ESR). Linear mixed models and Cox regression were used, stratified for autoantibody positivity, defined as IgG anti-CCP2 and/or IgM rheumatoid factor positivity. In total, 823 patients had autoantibody-positive RA (mean age 55 years, 67% female); 462 patients had autoantibody-negative RA (age 60 years, 64% female). Age, gender, and percentage of autoantibody-positive patients were stable throughout the inclusion periods. Disease activity significantly decreased over time within both groups. SDFR rates

**Data Availability Statement:** Our dataset, used for the analyses, contains potentially identifying and sensitive patient information. Therefore our data cannot be made publicly available because it is prohibited by Dutch law in the regulation "Algemene verordening gegevensbescherming" and does not comply with the study protocol as it was submitted to the local ethics committee. Inquiries can be directed to eac@lumc.nl.

**Funding:** The research leading to these results has received funding from the from the Dutch Arthritis Foundation and European Research Council (ERC, https://erc.europa.eu/) under the European Union's Horizon 2020 research and innovation programme (Starting grant, agreement No 714312, AHMvdHvM). The funding source had no role in the design and conduct of the study; collection, management, analysis, and interpretation of the data; preparation, review, or approval of the manuscript; or decision to submit the manuscript for publication.

**Competing interests:** I have read the journal's policy and the authors of this manuscript have the following competing interests: TH is a member of the Editorial Board of PLOS Medicine.

**Abbreviations:** ACPA, anti-citrullinated protein antibody; DAS, Disease Activity Score; DAS28-ESR, Disease Activity Score–28 with erythrocyte sedimentation rate; DMARD, disease-modifying antirheumatic drug; EAC, Early Arthritis Clinic; GP, general practitioner; HAQ, Health Assessment Questionnaire; HR, hazard ratio; RA, rheumatoid arthritis; RF, rheumatoid factor; SDFR, sustained DMARD-free remission.

increased after introduction of treat-to-target (hazard ratio [HR] 2006–2010 relative to 1993–1996: 3.35 [95% CI 1.46 to 7.72; $p = 0.004$]; HR 2011–2016: 4.57 [95% CI 1.80 to 11.6; $p = 0.001$]) in autoantibody-positive RA, but not in autoantibody-negative RA. In autoantibody-positive RA, mortality decreased significantly after the introduction of treat-to-target treatment adjustments (HR 2006–2010: 0.56 [95% CI 0.34 to 0.92; $p = 0.023$]; HR 2011–2016: 0.33 [95% CI 0.14 to 0.77; $p = 0.010$]), but not in autoantibody-negative RA (HR 2006–2010: 0.79 [95% CI 0.40 to 1.56; $p = 0.50$]; HR 2011–2016: 0.36 [95% CI 0.10 to 1.34; $p = 0.13$]). Similarly, functional disability improved in autoantibody-positive RA for the periods after 2000 relative to 1993–1996 (range −0.16 [95% CI −0.29 to −0.03; $p = 0.043$] to −0.32 [95% CI −0.44 to −0.20; $p < 0.001$] units of improvement), but not in autoantibody-negative RA (range 0.10 [95% CI −0.12 to 0.31; $p = 0.38$] to −0.13 [95% CI −0.34 to 0.07; $p = 0.20$] units of improvement). Limitations to note were that treatment was not randomized—but it was protocolized and instrumental variable analysis was used to obtain comparable groups—and that a limited spread of ethnicities was included.

## Conclusions

Although disease activity has improved in both autoantibody-positive and autoantibody-negative RA in recent decades, the response in long-term outcomes differed. We propose that it is time to subdivide RA into autoantibody-positive RA (type 1) and autoantibody-negative RA (type 2), in the hope that this leads to stratified treatment in RA.

## Author summary

### Why was this study done?

- Patients with rheumatoid arthritis (RA) have different risk factors and histology (microscopic anatomy) depending on the presence or absence of autoantibodies (anti-citrullinated protein antibodies and rheumatoid factor).

- Because it is suspected that RA with and without autoantibodies are 2 distinct diseases with different pathophysiology, we hypothesized that these 2 types of RA will have reacted differently to improvements in treatment strategies that have taken place over the last decades.

### What did the researchers do and find?

- Since its start in 1993, the inclusion criteria of the Leiden Early Arthritis Clinic cohort have not changed, and included RA patients have remained similar, apart from earlier diagnosis; therefore, RA patients from different years were comparable. Treatment protocols enhanced over time, but were similar for patients with and without autoantibodies.

- We studied the changes in disease activity and 3 long-term outcomes of RA patients with and without autoantibodies over time (inclusion period was a proxy for treatment strategy).

- We found that while disease activity improved in both patient groups, the long-term outcomes (the possibility to permanently stop medication, mortality, and functional disability) only improved in RA patients with autoantibodies.

### What do these findings mean?

- The disconnection between improvement in disease activity and subsequent improvement in long-term outcomes in RA without autoantibodies suggests that the underlying pathogenesis of RA with and without autoantibodies is different.
- We propose that it is time to formally subdivide RA into type 1 (with autoantibodies) and type 2 (without autoantibodies).

## Introduction

Careful clinical observations over time have led to the description of diseases. In addition, the subdividing of diseases has also been based on clinical observations, with differences in pathogenetic etiology identified subsequently. For instance, the subdividing of diabetes into type 1 and type 2 was based on differences in clinical presentation (young patients versus older and obese patients); this distinction was confirmed by treatment response to insulin, and subsequently fueled targeted etiological studies [1].

Rheumatoid arthritis (RA) is considered a syndrome. During the last decade it was observed that there are differences in RA patients with and without autoantibodies (such as rheumatoid factor [RF] and anti-citrullinated protein antibodies [ACPAs]). Autoantibody-positive RA has a different genetic background [2], different environmental risk factors [3,4], slight differences in the preclinical symptomatic phase and first clinical presentation [5–7], differences in histology [8], differences in the synovial fluid cytokine profile [9], and, when left untreated, more severe joint destruction [5]. Nonetheless, the etiology and pathophysiology of RA is still incompletely understood. It is unclear if there is 1 pathophysiological genesis—in which the presence of autoantibodies is promoted by certain genetic factors and where autoantibodies act as a "severity" factor—or, alternatively, if there are 2 different mechanisms of disease development. When distinct disease mechanisms exist, treatment response may differ. Whether autoantibody-positive and autoantibody-negative RA have different mechanisms can therefore be addressed by clinical evaluation of long-term results in response to changes in treatment strategy.

Slight differences in the effect of some drugs have been described between autoantibody-positive and autoantibody-negative RA patients based on trial data [10–13], but these are based on selected groups of RA patients with a limited follow-up duration. We will take advantage of a large longitudinal cohort including incident RA patients without selection followed over the last 25 years; to our knowledge, this is currently the largest observational cohort of RA. Treatment of RA has changed over time, and the same improvements in strategies (e.g., earlier treatment initiation and treat-to-target treatment adjustments) have been applied in both autoantibody-positive and autoantibody-negative patients. To evaluate whether autoantibody-positive RA and autoantibody-negative RA are 2 disease types, we studied the associations between changing treatment strategies and disease activity in the short term as well as 3 long-term outcomes.

## Methods

### Longitudinal cohort

The Leiden Early Arthritis Clinic (EAC) cohort is a population-based inception cohort including all consecutive patients newly presenting with recent-onset arthritis, that was started in 1993 and has been described in [14]. Inclusion criteria were presence of synovitis determined at physical examination by a rheumatologist and symptom duration of <2 years. The department of rheumatology in the Leiden University Medical Center is the only center for rheumatic diseases in a semi-rural area with >400,000 inhabitants. Since the start of the cohort, general practitioners (GPs) were informed on the relevance of early referral, and patients referred with suspicion of early arthritis were seen with priority, generally within 2 weeks. Of note, in line with Dutch GP guidelines, autoantibodies were rarely determined in primary care [15]. Written informed consent was obtained from all participants. The study was approved by the local medical ethics committee (Commissie Medische Ethiek of the Leiden University Medical Center; B19.008).

For this study we selected patients with RA (clinical diagnosis plus fulfillment of 1987 American College of Rheumatology criteria). The use of the 1987 criteria (instead of the 2010 criteria) excluded influences of temporal changes in views on diagnosing RA and of the inverse relationship between presence of autoantibodies and degree of inflammation on the classification [16,17]. Between 24 February 1993 and 31 December 2016, 1,377 patients enrolled in the cohort were classified with RA.

At the first visit, rheumatologists and patients completed questionnaires (including the Health Assessment Questionnaire [HAQ] Disability Index), swollen and tender joint counts were performed, and blood samples were taken for routine diagnostic laboratory screening (including erythrocyte sedimentation rate [ESR] and immunoglobulin M RF [positive if ≥3.5 IU/ml]). From 2006 onward, ACPA was measured (before 2009, anti-CCP2, Eurodiagnostica, positive if ≥25 U/ml; from 2009 onward, EliA, Phadia, positive if ≥7U/ml). In patients included before 2006, ACPA status was assessed retrospectively on stored baseline serum samples using the Eurodiagnostica assay. Since seroconversion is rare, repeated ACPA and/or RF measurements during follow-up were not studied [18]. In 6 patients, autoantibody status was not available; consequently, they were excluded from the analyses (S1 Fig).

Protocolized follow-up visits were performed twice in the first year and yearly thereafter, as long as patients were treated at the outpatient clinic. Follow-up ended in case of death, release from care due to sustained DMARD-free remission (SDFR), moving to another area, or withdrawal of informed consent while remaining treated. As data were collected at regular rheumatologist visits, withdrawal of informed consent was rare. Data from Statistics Netherlands from our region showed that moving away from the Leiden area was also infrequent (<3% annually) [19]. Inherent to the design, follow-up was shorter in the more recent inclusion periods. The majority of missed follow-up visits (not due to inclusion date) was due to mortality or SDFR.

### Definition of autoantibody-positive and autoantibody-negative

Patients with ACPA and/or RF were categorized as autoantibody-positive (type 1); double-negative patients were categorized as autoantibody-negative (type 2). For practical reasons, the distinction between type 1 and type 2 is based on the autoantibodies that are currently used in the clinic. It could be that if more factors were included, e.g., other autoantibodies or other factors such as obtained from histology, a better division into groups would be obtained [20–23]. Our primary goal, however, was to investigate the main distinction into autoantibody-positive and autoantibody-negative RA as it is used in clinical practice.

## Treatment

Patients were treated in routine care according to protocols. Of 1,377 RA patients, 86 were treated within randomized clinical trials that were not in line with the treatment guidelines at that time and were excluded, leaving 1,285 RA patients for analyses (S1 Fig). Temporal changes in treatment strategies concerned the initial start as well as treatment adjustments over time; improvements in both aspects of treatment are reflected by inclusion period as proxy. Patients included between 24 February 1993 and 31 December 1996 ($n$ = 168) received initial nonsteroidal anti-inflammatory drugs (NSAIDs) and started mild disease-modifying antirheumatic drugs (DMARDs) with delay. Patients included between 1 January 1997 and 31 December 2000 ($n$ = 185) were treated early with DMARDs but not with methotrexate (e.g., hydroxychloroquine and sulfasalazine) [24]. Patients included between 1 January 2001 and 31 December 2005 ($n$ = 207) started early with methotrexate [25]. From 2006 onwards, early methotrexate was followed by treat-to-target treatment adjustments, indicating treatment adjustments in case of increased Disease Activity Score (DAS) (1 January 2006–31 December 2010, $n$ = 335) [26]. Furthermore, because the value of very early treatment became even more apparent in 2010, and as GP delay contributed most to the total delay in our region [27], from 2011 onwards, on top of the existing regimen, additional efforts were undertaken to further reduce referral delay by instituting an early arthritis recognition clinic, which is a screening clinic for the presence of inflammatory arthritis (1 January 2011–31 December 2016, $n$ = 390) [27–29].

In line with the absence of guidelines that initial treatment should be adapted to autoantibody status [30,31], initial treatment choices were not directed by autoantibodies. Subsequent treatment decisions were targeted at DAS; this was independent of patient characteristics. Thus, protocols were similar for type 1 and 2.

Anti-TNF was the first biologic that became available in the early 2000s for RA patients whose treatment failed on ≥2 conventional DMARDs [32]. Over time other biologics were registered, though the indication remained similar in the Netherlands. S1 Table provides information about the use of biologics during different follow-up durations, for type 1 and 2 separately. The usage was slightly higher in type 1, especially after introduction of treat-to-target.

## Outcomes

Disease activity reflected the direct results of treatment, as measured with the Disease Activity Score–28 with erythrocyte sedimentation rate (DAS28-ESR) [33]. Since 2006, treatment has been aimed at this short-term target to eventually improve long-term outcomes. Three long-term outcomes were studied: SDFR, mortality, and functional disability. SDFR was defined as the sustained absence of synovitis (by physical examination) after discontinuation of DMARD therapy (including biologics and systemic or intra-articular corticosteroids) for the entire follow-up after DMARD withdrawal, and this follow-up had to continue for at least 1 year after DMARD stop [34]. This stringent and innovative definition of long-term remission is the opposite of disease persistence and has become increasingly achievable [35]. After achievement of SDFR, patients were followed for median 5.5 years, to verify its sustainability. Patients who achieved DMARD-free remission but developed a late flare during subsequent follow-up ($n$ = 23) were not considered as being in SDFR. All medical files of patients with ≥1 year of follow-up were retrospectively explored for SDFR until April 2017. Mortality status was obtained from the civic registries on 1 June 2018. Functional disability is one of the most important outcomes from the patients' perspective [36], and was measured yearly with the HAQ, ranging from 0 (no disability) to 3 (severe disability) [37,38].

## Statistical analyses

Main analyses were done for type 1 and 2 RA separately. Inclusion period was used as an instrumental variable for treatment strategy. Within each type, improvements over time were compared to the reference period (inclusion in 1993–1996).

Next, improvements over time compared to the reference period were compared between the 2 types by including an interaction term in the models to quantify the difference in improvement over time between the 2 types.

Time to SDFR was analyzed with Cox regression. SDFR status was censored at the date of data extraction (e.g., revision of the medical files) or at an earlier date when patients were lost to follow-up or had died.

Mortality was analyzed with Cox regression; follow-up was censored at the date of data extraction. Mortality was not compared to the general population because excess mortality in RA relative to the general population requires >10 years of follow-up to become apparent [39,40]; this follow-up duration was absent for the recent inclusion periods.

Missing data on DAS28-ESR (complete DAS28-ESR missing for 0% of patients at baseline and 3% of patients at follow-up visits) and HAQ (missing for 13% at baseline and 22% at annual follow-up visits) for attended visits were imputed using multivariate multiple imputation with predictive mean matching (100 cycles, 30 datasets). DAS28-ESR and HAQ were analyzed with linear mixed models. Because both outcomes rapidly decreased within the first year, the first year was analyzed separately from the remaining follow-up [41–43]. The slope of decrease in the first year was analyzed with a random intercept and an identity covariance matrix. The course after the first year was analyzed with a random intercept, random slope, and continuous auto-regressive covariance matrix of order 1. Estimated marginal means were calculated. Percentages of DAS28-ESR remission (<2.6) at 1 and 3 years were tested with chi-squared tests [44].

To minimize the influence of the association of the studied exposure and follow-up duration, analyses were truncated at 15 years of follow-up, and follow-up duration was not included as a covariate in any of the analyses. All analyses were corrected for age and gender to improve model fit. As none of the measured baseline covariates were true confounders of the relationship between treatment strategy and outcomes, because they were not associated with the exposure or regarded to be in the causal path (see S1 Text and S2 Fig for explanation), no other corrections were made.

No formal prospective analysis plan was written or submitted prior to performing the analyses. The widths of the confidence intervals have not been adjusted for multiplicity, and p-values < 0.05 were considered significant. R 3.6.1 with packages described in S2 Text was used. This study is reported as per the Strengthening the Reporting of Observational Studies in Epidemiology (STROBE) guideline (See S1 Checklist).

## Sensitivity analyses

In a sensitivity analysis RA was defined according to the 2010 criteria.

In response to requests during peer review, to assess whether the difference in age at onset between the disease types might influence the results, patients aged <65 years at diagnosis were analyzed in a sensitivity analysis.

For SDFR and mortality, a sensitivity analysis was done to address differences in symptom duration at baseline, as patients could not have presented themselves to the EAC if the studied event (SDFR or death) had already happened. To assess the influence of this possible left truncation, correction for left truncation was applied.

Finally, data for both disease types were plotted by inclusion period for all outcomes; this was done for illustration.

## Results

### Baseline characteristics

In total, 823 patients had type 1 RA; their mean age at first presentation was 55 years, and 67% were female (Table 1). In total, 462 patients had type 2 RA; their mean age at first presentation was 60 years, and 64% were female. Age, gender, and percentage of RA types were stable throughout the inclusion periods ($p = 0.59$, $p = 0.28$, and $p = 0.42$, respectively), showing that similar RA patients were included over time. Within both RA types, patients presented with shorter symptom duration, lower numbers of swollen and tender joints, and lower acute phase reactants in more recent inclusion periods, reflecting that earlier presentation was paralleled with less severe disease (Table 1).

**Disease activity.** In type 1 RA, DAS28-ESR improved in the first year and during subsequent follow-up (Fig 1; Table 2). The percentage of patients achieving DAS28-ESR remission (<2.6) significantly increased, e.g., from 13% in the oldest inclusion period to 50% at year 1 and 61% at year 3 in the most recent period (S3 Fig).

**Table 1. Characteristics of patients with type 1 (autoantibody-positive) and type 2 (autoantibody-negative) RA at first presentation to the Leiden Early Arthritis Clinic.**

| Characteristic | Inclusion period | | | | | p-Value |
|---|---|---|---|---|---|---|
| | 1993–1996 | 1997–2000 | 2001–2005 | 2006–2010 | 2011–2016 | |
| **Type 1 RA** | (n = 112, 67%) | (n = 118, 64%) | (n = 129, 62%) | (n = 203, 61%) | (n = 261, 67%) | |
| Women, n (%) | 77 (69) | 82 (70) | 91 (71) | 136 (67) | 167 (64) | 0.70 |
| Age in years, mean (SD) | 56 (16) | 55 (16) | 55 (15) | 54 (15) | 56 (15) | 0.63 |
| Symptom duration, days, median (IQR) | 153 (84–306) | 156 (84–304) | 147 (72–264) | 146 (61–270) | 103 (53–227) | 0.006 |
| Current smoker, n (%) | 35 (33) | 35 (33) | 29 (27) | 40 (22) | 74 (30) | 0.21 |
| 28 SJC, median (IQR) | 6 (3–10) | 7 (4–12) | 4 (2–7) | 4 (2–7) | 4 (2–7) | <0.001 |
| 28 TJC, median (IQR) | 7 (3–13) | 7 (3–14) | 7 (3–12) | 6 (3–11) | 5 (2–9) | <0.001 |
| ESR, median (IQR) | 46 (26–70) | 32 (20–54) | 30 (18–55) | 29 (14–42) | 29 (14–41) | <0.001 |
| VAS general health, median (IQR) | 43 (17–70) | 44 (26–66) | 53 (34–72) | 56 (29–72) | 70 (50–80) | <0.001 |
| DAS28-ESR, median (IQR) | 5.5 (4.2–6.5) | 5.2 (4.2–6.1) | 5.2 (4.3–6.0) | 4.9 (4.2–6.0) | 4.8 (4.1–5.7) | 0.02 |
| HAQ, median (IQR) | 1.0 (0.6–1.4) | 0.8 (0.4–1.6) | 1.0 (0.6–1.6) | 1.0 (0.5–1.5) | 1.0 (0.5–1.5) | 0.12 |
| **Type 2 RA** | (n = 56, 33%) | (n = 67, 36%) | (n = 78, 38%) | (n = 132, 39%) | (n = 129, 33%) | |
| Women, n (%) | 38 (68) | 41 (61) | 57 (73) | 80 (61) | 79 (61) | 0.34 |
| Age in years, mean (SD) | 56 (15) | 59 (19) | 60 (14) | 61 (16) | 62 (14) | 0.16 |
| Symptom duration, days, median (IQR) | 126 (61–220) | 92 (62–219) | 120 (74–234) | 109 (59–176) | 85 (45–189) | 0.06 |
| Current smoker, n (%) | 17 (30) | 11 (18) | 14 (20) | 24 (21) | 28 (22) | 0.52 |
| 28 SJC, median (IQR) | 9 (4–14) | 12 (7–19) | 6 (3–10) | 6 (3–10) | 6 (3–10) | <0.001 |
| 28 TJC, median (IQR) | 9 (3–19) | 13 (6–20) | 11 (5–19) | 9 (4–13) | 7 (3–11) | <0.001 |
| ESR, median (IQR) | 40 (22–56) | 28 (16–47) | 27 (16–47) | 31 (9–46) | 25 (11–41) | 0.008 |
| VAS general health, median (IQR) | 46 (25–63) | 50 (26–62) | 56 (36–75) | 64 (44–79) | 70 (60–80) | <0.001 |
| DAS28-ESR, median (IQR) | 5.6 (4.5–6.3) | 5.8 (4.8–6.5) | 5.6 (4.4–6.7) | 5.3 (4.4–6.3) | 5.2 (4.4–6.0) | 0.19 |
| HAQ, median (IQR) | 1.1 (0.8–1.6) | 0.9 (0.5–1.4) | 1.1 (0.8–1.8) | 1.1 (0.8–1.5) | 1.0 (0.6–1.5) | 0.15 |

p-Value for results of Kruskal–Wallis H test (Fisher's exact test for proportions and ANOVA for normally distributed variables). The percentage of patients with type 1 or 2 RA for the different inclusion periods was stable over time ($p = 0.42$). SJC and TJC are the number of swollen and tender joints, respectively, out of 28 joints assessed. The VAS general health is a self-reported assessment, ranging from 0 to 100. DAS28-ESR ranges from 2 to 9.4, with higher scores indicating more disease activity. HAQ (HAQ Disability Index) ranges from 0 to 3, with higher scores indicating more disability.

DAS28-ESR, Disease Activity Score–28 with erythrocyte sedimentation rate; ESR, erythrocyte sedimentation rate; HAQ, Health Assessment Questionnaire; IQR, interquartile range; RA, rheumatoid arthritis; SD, standard deviation; SJC, swollen joint count; TJC, tender joint count; VAS, visual analogue scale.

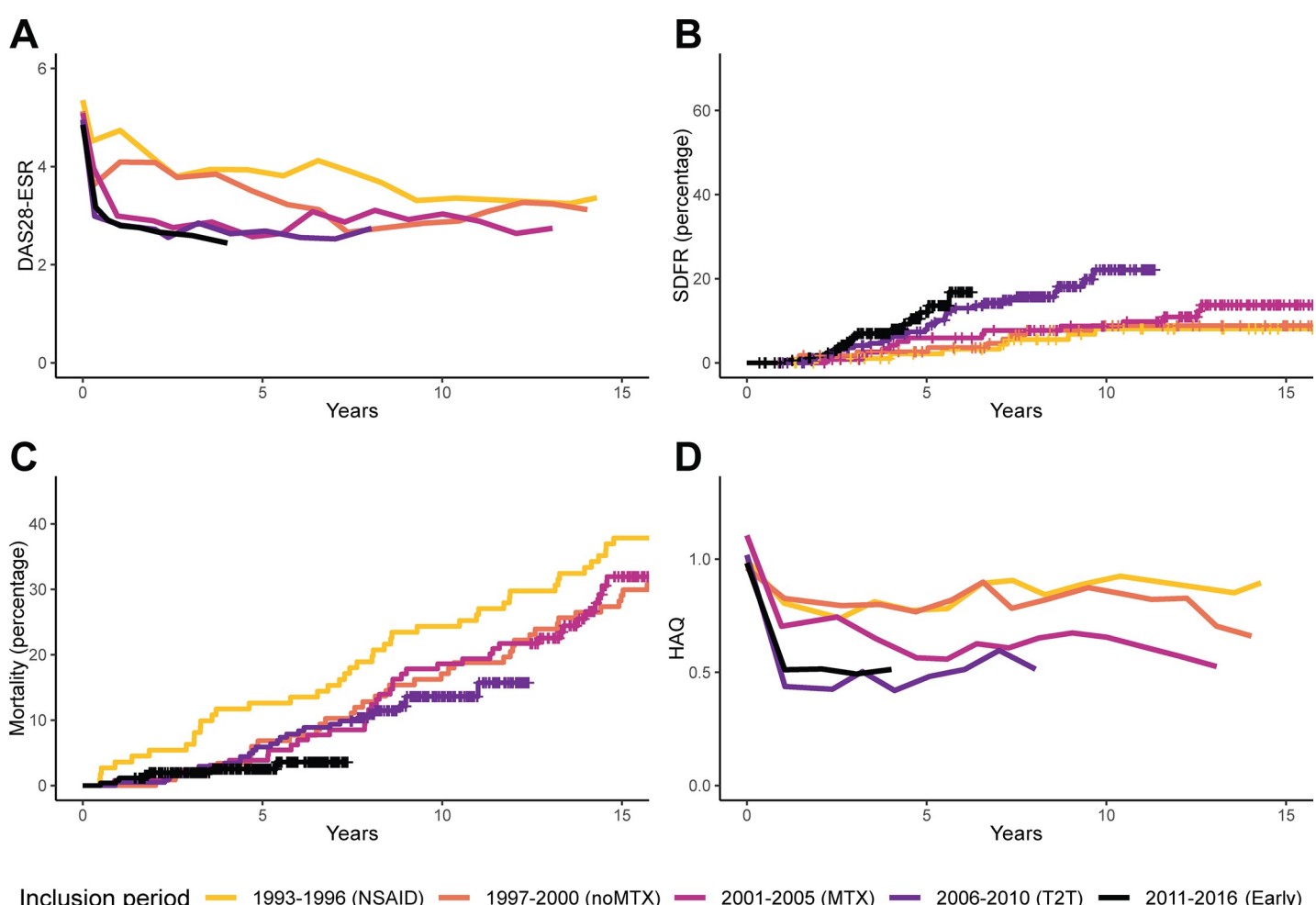

**Fig 1. Disease activity and long-term outcomes in type 1 RA.** Disease activity over time (A) and the long-term outcomes SDFR (B), mortality (C), and functional disability (D) in type 1 (autoantibody-positive) RA. For DAS28-ESR and HAQ, mean values of imputed data from visits that were attended are shown; when <20% of patients attended the visit, lines were truncated. DAS28-ESR ranges from 2 to 9.4, with higher scores indicating more disease activity. Remission is defined as a score < 2.6, and a change of >1.2 is considered a clinically relevant change [44]. HAQ ranges from 0 to 3, with higher scores indicating more disability. The minimally important difference is 0.22 [38]. For SDFR, at 5 years, 85%, 87%, 89%, 82%, and 32% of patients from inclusion period 1993–1996 to 2011–2016, respectively, were still at risk. At 10 years, the proportion at risk was 79%, 71%, 70%, 15%, and 0%, and at 15 years, 56%, 59%, 12%, 0%, and 0%. For mortality, at 5 years, 87%, 93%, 96%, 94%, and 42% of patients from inclusion period 1993–1996 to 2011–2016, respectively, were still at risk. At 10 years, the proportion at risk was 76%, 83%, 81%, 38%, and 0%, and at 15 years, 62%, 71%, 35%, 0%, and 0%. DAS28-ESR, Disease Activity Score–28 with erythrocyte sedimentation rate; Early, early treatment; HAQ, Health Assessment Questionnaire; NSAID, nonsteroidal anti-inflammatory drug; MTX, methotrexate; noMTX, no methotrexate; RA, rheumatoid arthritis; SDFR, sustained DMARD-free remission; T2T, treat-to-target.

In type 2 RA, DAS28-ESR also improved, especially in the first year (Fig 2; Table 3). DAS28-ESR remission percentages increased from 32% in the oldest inclusion period to 54% at year 1 and 71% at year 3 in the most recent period (S3 Fig).

**Sustained DMARD-free remission.** In type 1 RA, SDFR significantly increased over time, especially since the start of treat-to-target (Fig 1; Table 2). In type 2 RA, there was no significant increase in SDFR (Fig 2; Table 3).

**Mortality.** Compared to the reference period, mortality decreased significantly in type 1 RA since the start of treat-to-target (Fig 1; Table 2). No significant association was found in type 2 RA (Fig 2; Table 3), although hazard ratios were in the same direction as in type 1 RA.

**Functional disability.** In type 1 RA, functional disability improved over time since the start of early methotrexate as the standard treatment, both in the first year and the

**Table 2. Disease activity during the first year and subsequent follow-up, and long-term outcomes (sustained DMARD-free remission, mortality, and functional disability) by inclusion period compared to the reference period for type 1 (autoantibody-positive) RA.**

| Inclusion period | DAS28-ESR, slope in first year | | DAS28-ESR over time after first year | | Sustained DMARD free remission | | Mortality | | HAQ, slope in first year | | HAQ over time, after first year | |
|---|---|---|---|---|---|---|---|---|---|---|---|---|
| | Relative mean difference[a] | p-val | Relative mean difference[b] | p-val | Hazard ratio[c] | p-val | Hazard ratio[c] | p-val | Relative mean difference[a] | p-val | Relative mean difference[b] | p-val |
| 1993–1996 | Ref[d] | | Ref[d] | | Ref | | Ref | | Ref[d] | | Ref[d] | |
| 1997–2000 | −0.38 (−0.87; 0.10) | 0.12 | **−0.41 (−0.66; −0.16)** | **0.002** | 1.14 (0.42; 3.05) | 0.80 | 0.74 (0.47; 1.15) | 0.18 | 0.01 (−0.19; 0.21) | 0.89 | −0.02 (−0.15; 0.11) | 0.58 |
| 2001–2005 | **−1.70 (−2.21; −1.20)** | **<0.001** | **−0.86 (−1.12; −0.61)** | **<0.001** | 1.66 (0.67; 4.12) | 0.27 | 0.71 (0.46; 1.11) | 0.13 | **−0.28 (−0.49; −0.07)** | **0.009** | **−0.16 (−0.29; −0.03)** | **0.043** |
| 2006–2010 | **−1.62 (−2.08; −1.17)** | **<0.001** | **−1.04 (−1.28; −0.80)** | **<0.001** | **3.35 (1.46; 7.72)** | **0.004** | **0.56 (0.34; 0.92)** | **0.023** | **−0.33 (−0.51; −0.14)** | **0.001** | **−0.32 (−0.44; −0.20)** | **<0.001** |
| 2011–2016 | **−1.54 (−1.96; −1.12)** | **<0.001** | **−1.07 (−1.32; −0.83)** | **<0.001** | **4.57 (1.80; 11.6)** | **0.001** | **0.33 (0.14; 0.77)** | **0.010** | **−0.29 (−0.46; −0.12)** | **0.001** | **−0.26 (−0.38; −0.14)** | **0.008** |

Bold value indicate p-values < 0.05.

[a]Difference in slope in the first year in the inclusion period compared to 1993–1996; analyzed with linear mixed models corrected for age and gender. A negative number indicates a steeper slope.

[b]Difference in mean over time in the inclusion period compared to 1993–1996; analyzed with linear mixed models corrected for age and gender.

[c]Hazard ratios compared to 1993–1996; analyzed with Cox regression corrected for age and gender.

[d]The estimated marginal mean, adjusted for age and gender, in type 1 RA for inclusion period 1993–1996 was −0.34 (95% CI −0.70 to 0.03) for the slope in DAS28-ESR in the first year, 3.58 (95% CI 3.39 to 3.76) for DAS28-ESR over time after the first year, −0.15 (95% CI −0.29 to 0.00) for slope in HAQ in the first year, and 0.78 (95% CI 0.68 to 0.88) for HAQ over time after the first year.

DAS28-ESR, Disease Activity Score–28 with erythrocyte sedimentation rate; DMARD, disease-modifying antirheumatic drug; HAQ, Health Assessment Questionnaire; p-val, p-value.

subsequent years (Fig 1; Table 2). In type 2, in contrast, improvement was absent (Fig 2; Table 3).

## Comparison of improvement of type 1 and type 2

To assess whether more improvement was indeed observed in type 1 RA compared to type 2 RA, change with respect to the reference period was compared between the 2 disease types by adding an interaction term to the models. More improvement for the outcomes DAS28-ESR over time, SDFR, and functional disability was observed in type 1 RA (Table 4). This improvement was statistically significant for these outcomes in the inclusion period 2006–2010 (early methotrexate followed by treat-to-target treatment adjustments).

## Sensitivity analyses

According to the 2010 criteria, 1,421 patients had RA, 957 type 1 and 474 type 2 (S4 Fig). Due to the composition of these criteria, type 2 RA required ≥11 involved joints for classification [16,17]. Indeed, this group had high joint counts, especially high tender joints in the latest periods, when acute phase reactants and swollen joint counts at diagnosis decreased (S2 Table). This possibly resulted in incomparability in disease activity between the periods within type 2 RA. Results for type 1 classified by the 2010 criteria were similar to those when RA was classified according to the 1987 criteria. For type 2 little improvement in DAS28-ESR was present, and effect sizes of long-term outcomes were in line with the main results (S3 and S4 Tables).

Analyses were repeated in patients aged <65 years at diagnosis; similar results were obtained except for a non-significant improvement in mortality in type 1 RA, possibly caused by a lower number of events (S5 and S6 Tables).

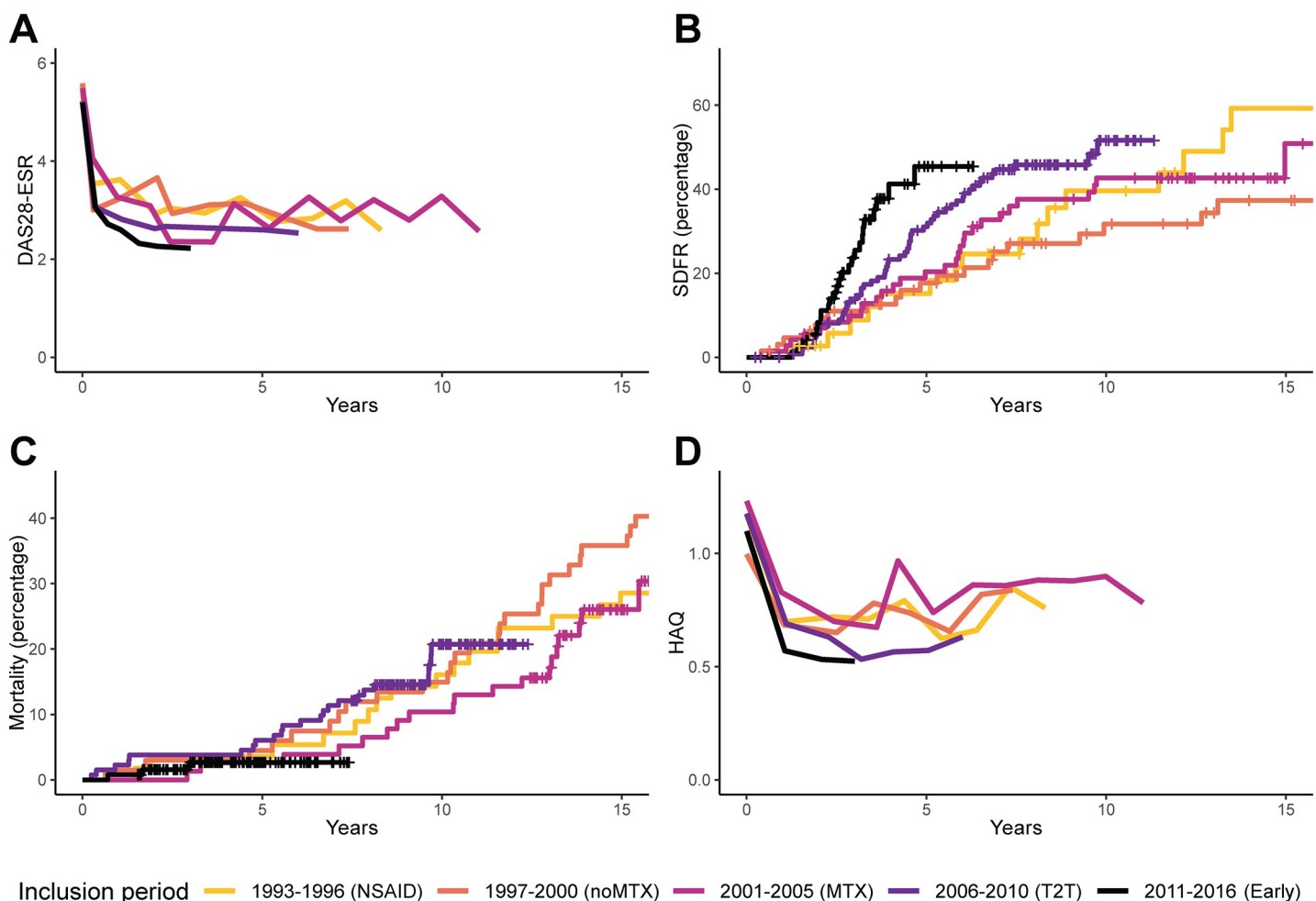

**Fig 2. Disease activity and long-term outcomes in type 2 RA.** Disease activity over time (A) and the long-term outcomes SDFR (B), mortality (C), and functional disability (D) in type 2 (autoantibody-negative) RA. For DAS28-ESR and HAQ, mean values of imputed data from visits that were attended are shown; when <20% of patients attended the visit, lines were truncated. DAS28-ESR ranges from 2 to 9.4, with higher scores indicating more disease activity. Remission is defined as a score < 2.6, and a change of >1.2 is considered a clinically relevant change [44]. HAQ ranges from 0 to 3, with higher scores indicating more disability. The minimally important difference is 0.22 [38]. For SDFR, at 5 years, 73%, 74%, 72%, 62%, and 14% of patients from inclusion period 1993–1996 to 2011–2016, respectively, were still at risk. At 10 years, the proportion at risk was 41%, 45%, 47%, 9%, and 0%, and at 15 years, 22%, 31%, 8%, 0%, and 0%. For mortality, at 5 years, 96%, 96%, 97%, 94%, and 27% of patients from inclusion period 1993–1996 to 2011–2016, respectively, were still at risk. At 10 years, the proportion at risk was 84%, 85%, 90%, 34%, and 0%, and at 15 years, 71%, 64%, 26%, 0%, and 0%. DAS28-ESR, Disease Activity Score–28 with erythrocyte sedimentation rate; Early, early treatment; HAQ, Health Assessment Questionnaire; NSAID, nonsteroidal anti-inflammatory drug; MTX, methotrexate; noMTX, no methotrexate; RA, rheumatoid arthritis; SDFR, sustained DMARD-free remission; T2T, treat-to-target.

Effect sizes for the outcomes SDFR and mortality after correction for left truncation were similar (S7 Table).

For illustration, head-to-head comparisons between type 1 and type 2 RA within the inclusion periods are shown in S5–S8 Figs.

## Discussion

### Summary of findings

During the last 25 years, the treatment of RA has changed in several aspects. We studied outcomes of RA and observed that improved treatment strategies were paralleled by reduced disease activity in autoantibody-positive and autoantibody-negative RA, but resulting significant

**Table 3. Disease activity during the first year and subsequent follow-up, and long-term outcomes (sustained DMARD-free remission, mortality, and functional disability) by inclusion period compared to the reference period for type 2 (autoantibody-negative) RA.**

| Inclusion period | DAS28-ESR, slope in first year | | DAS28-ESR over time after first year | | Sustained DMARD free remission | | Mortality | | HAQ, slope in first year | | HAQ over time, after first year | |
|---|---|---|---|---|---|---|---|---|---|---|---|---|
| | Relative mean difference[a] | p-val | Relative mean difference[b] | p-val | Hazard ratio[c] | p-val | Hazard ratio[c] | p-val | Relative mean difference[a] | p-val | Relative mean difference[b] | p-val |
| 1993–1996 | Ref[d] | | Ref[d] | | Ref | | Ref | | Ref[d] | | Ref[d] | |
| 1997–2000 | −0.53 (−1.30; 0.24) | 0.18 | 0.08 (−0.32; 0.49) | 0.69 | 0.61 (0.32; 1.18) | 0.14 | 0.67 (0.35; 1.30) | 0.24 | 0.16 (−0.13; 0.44) | 0.29 | 0.03 (−0.19; 0.24) | 0.81 |
| 2001–2005 | **−0.88 (−1.66; −0.11)** | **0.025** | −0.03 (−0.43; 0.37) | 0.89 | 0.80 (0.43; 1.48) | 0.48 | 0.57 (0.28; 1.13) | 0.11 | 0.05 (−0.25; 0.35) | 0.75 | 0.10 (−0.12; 0.31) | 0.38 |
| 2006–2010 | **−0.78 (−1.48; −0.08)** | **0.029** | −0.26 (−0.63; 0.11) | 0.17 | 1.11 (0.63; 1.97) | 0.71 | 0.79 (0.40; 1.56) | 0.50 | 0.02 (−0.24; 0.28) | 0.87 | −0.09 (−0.28; 0.10) | 0.34 |
| 2011–2016 | **−1.08 (−1.75; −0.41)** | **0.002** | **−0.44 (−0.84; −0.04)** | **0.030** | 1.89 (0.97; 3.67) | 0.060 | 0.36 (0.10; 1.34) | 0.13 | −0.02 (−0.27; 0.23) | 0.89 | −0.13 (−0.34; 0.07) | 0.20 |

Bold values indicate p-values < 0.05.

[a]Difference in slope in the first year in the inclusion period compared to 1993–1996; analyzed with linear mixed models corrected for age and gender. A negative number indicates a steeper slope.

[b]Difference in mean over time in the inclusion period compared to 1993–1996; analyzed with linear mixed models corrected for age and gender.

[c]Hazard ratios compared to 1993–1996; analyzed with Cox regression and corrected for age and gender.

[d]The estimated marginal mean, adjusted for age and gender, in type 2 RA for inclusion period 1993–1996 was −1.27 (95% CI −1.81 to −0.72) for the slope in DAS28-ESR in the first year, 2.70 (95% CI 2.40 to 3.01) for DAS28-ESR over time after the first year, −0.46 (95% CI −0.67 to −0.25) for slope in HAQ in the first year, and 0.62 (95% CI 0.47 to 0.78) for HAQ over time after the first year.

DAS28-ESR, Disease Activity Score–28 with erythrocyte sedimentation rate; DMARD, disease-modifying antirheumatic drug; HAQ, Health Assessment Questionnaire; p-val, p-value.

improvements in long-term outcomes—SDFR, mortality, and functional disability—were only present in autoantibody-positive RA and not in autoantibody-negative RA. In line with these findings, DAS28-ESR, SDFR, and functionality showed greater improvements over the last 25 years within autoantibody-positive than within autoantibody-negative RA. Especially the introduction of treat-to-target treatment adjustments was associated with significantly greater improvements in autoantibody-positive RA than in autoantibody-negative RA. The disconnection between improvements in disease activity and in several long-term outcomes suggests that the underlying pathogenesis of autoantibody-positive and autoantibody-negative RA is different. We therefore propose that the time has come to subdivide RA into type 1 and type 2.

## Comparisons with other studies

Subdivisions of disease are ideally underpinned with identified differences in etiopathology. However, clinical observations have frequently been the basis of subdivisions of diseases and have preceded the identification of pathophysiological mechanisms. Both types of RA have a different genetic background. Whereas >100 genetic risk factors are identified for type 1, few genetic factors have been related to type 2 RA [45]. Known environmental risk factors are associated with predominantly 1 of the 2 types [3,4]. These data, together with observed differences in histology [8], may also point towards different underlying mechanisms.

Etiopathogenetic research in the last decade has focused mostly on autoantibody-positive RA, but a causal relationship for the autoantibodies has not been proven. Further pathogenic research is needed for both type 1 and type 2 RA.

**Table 4. Differences in improvement of disease outcomes between type 1 (autoantibody-positive) and type 2 (autoantibody-negative) rheumatoid arthritis with enhanced treatment strategies over 25 years.**

| Inclusion period | DAS28-ESR, slope in first year | | DAS28-ESR over time after first year | | Sustained DMARD free remission | | Mortality | | HAQ, slope in first year | | HAQ over time, after first year | |
|---|---|---|---|---|---|---|---|---|---|---|---|---|
| | Relative mean difference[a] | p-val | Relative mean difference[b] | p-val | Hazard ratio[c] | p-val | Hazard ratio[d] | p-val | Relative mean difference[a] | p-val | Relative mean difference[b] | p-val |
| 1993–1996 | Ref[d] | | Ref[d] | | Ref | | Ref | | Ref[d] | | Ref[d] | |
| 1997–2000 | 0.14 (−0.75; 1.04) | 0.75 | −0.46 (−0.94; 0.03) | 0.068 | 1.80 (0.55; 5.92) | 0.33 | 1.02 (0.47; 2.23) | 0.96 | −0.14 (−0.49; 0.21) | 0.42 | −0.06 (−0.30; 0.19) | 0.65 |
| 2001–2005 | −0.82 (−1.73; 0.08) | 0.073 | **−0.70 (−1.18; −0.22)** | **0.004** | 2.10 (0.70; 6.28) | 0.18 | 1.22 (0.54; 2.73) | 0.64 | −0.33 (−0.69; 0.03) | 0.069 | −0.21 (−0.46; 0.04) | 0.095 |
| 2006–2010 | −0.82 (−1.64; 0.00) | 0.050 | **−0.70 (−1.14; −0.25)** | **0.002** | **2.93 (1.08; 7.90)** | **0.034** | 0.82 (0.37; 1.83) | 0.63 | **−0.35 (−0.66; −0.05)** | **0.024** | **−0.22 (−0.44; 0.00)** | **0.046** |
| 2011–2016 | −0.47 (−1.23; 0.29) | 0.22 | **−0.55 (−1.04; −0.05)** | **0.030** | 2.10 (0.71; 6.22) | 0.18 | 1.11 (0.26; 4.85) | 0.89 | −0.27 (−0.56; 0.02) | 0.064 | −0.11 (−0.35; 0.13) | 0.37 |

Bold values indicate p-values < 0.05. The overall p-value of the interaction term in the models (e.g., the p-value for difference in improvement between the 2 subtypes over all inclusion periods) was 0.072 for DAS28-ESR slope in first year, <0.001 for DAS28-ESR over time after first year, 0.28 for sustained DMARD-free remission, 0.91 for mortality, 0.016 for HAQ slope in first year, and 0.10 for HAQ over time after first year.

[a]Additional improvement in type 1 relative to type 2. A negative number corresponds to additional change downward in type 1 relative to the reference period (e.g., more decrease in the first year with respect to the reference period). Since lower DAS28-ESR/HAQ is better, a negative number indicates more improvement in type 1.

[b]Additional improvement in type 1 relative to type 2. A negative number corresponds to additional change downward of the mean after the first year in type 1 relative to the reference period. Since lower DAS28-ESR/HAQ is better, a negative number indicates more improvement in type 1.

[c]Additional improvement in type 1 relative to type 2. A number above 1 corresponds to additional SDFR in type 1 relative to the reference period. Since more SDFR is better, a number above 1 indicates more improvement in type 1.

[d]Additional improvement in type 1 relative to type 2. A number below 1 corresponds to less mortality in type 1 relative to the reference period. Since lower mortality is better, a number below 1 indicates more improvement in type 1.

DAS28-ESR, Disease Activity Score–28 with erythrocyte sedimentation rate; DMARD, disease-modifying antirheumatic drug; HAQ, Health Assessment Questionnaire

p-val, p-value.

## Strengths and limitations of this study

We have studied the autoantibodies that are in daily use in clinical practice (ACPA and RF). Several new autoantibodies have recently been identified; most co-occur in patients who also harbor ACPA or RF [20–23]. A small percent of ACPA- and RF-negative patients were found to be positive for novel autoantibodies, leaving the so-called serological gap largely unchanged. There was insufficient power to assess which autoantibodies are optimal for the characterization of type 1 RA. It is a subject for further research to determine whether the division can be optimized by incorporation of recently identified autoantibodies or other markers (e.g., obtained from histology) [46].

Autoantibody positivity was determined with the cutoffs that are also used in daily clinical practice in our hospital. Some patients might have values just around the cutoff at baseline and therefore might change in autoantibody positivity over time. Previous research in the EAC cohort has shown that seroconversion towards autoantibody negativity is rare, even when SDFR is achieved, and that seroconversion was mostly caused by fluctuations of levels around the cutoff [18]. Similarly, data from our cohort show that seroconversion from autoantibody negativity to autoantibody positivity is also infrequent (2% after 1 year of follow-up; S9 Fig). Thus, autoantibody status is quite stable after diagnosis.

Patients with type 2 RA had a clinical diagnosis of RA, fulfilled classification criteria, and lacked ACPA and RF. It has been suggested that autoantibody-negative RA is heterogeneous in nature. We find it important to formally consider autoantibody-negative RA as a separate entity, but we cannot exclude the possibility that type 2 RA consists of different subtypes. This is beyond the scope and power of this study.

To assess the response to improved treatment strategies without exposing patients to out-dated and less effective treatments, historical data were used, and inclusion period was used as an instrumental variable for treatment strategy. As an alternative to randomization, instrumental variable analysis uses a proxy (inclusion period) to create groups with comparable patients that receive different treatment strategies. Between these groups, treatment strategies can be compared without confounding by indication, under the assumption that allocation to the groups is random. Since the inclusion criteria of the Leiden EAC have not changed over time, year of RA diagnosis was assumed random. Importantly, initial treatment protocols and treat-to-target protocols were similar for patients with and without autoantibodies, making the instrument similar for both patient groups.

Treatment was targeted at DAS remission since 2006, and was never targeted at autoantibodies (notably, ACPA results became available for rheumatologists in this study from 2006 onwards). While type 2 RA had a slightly higher baseline DAS28-ESR and in type 1 mean DAS28-ESR over time decreased more, mean DAS28-ESR and remission rates were similar or better in type 2 RA in all periods. Observed differences in long-term outcomes are therefore unlikely to be the result of better adherence to treat-to-target in autoantibody-positive patients. Also, the finding that patients with autoantibodies more often required biologics to achieve DAS28-ESR remission (S1 Table) merely underlines the difference between the types.

Progression of joint destruction was not studied as outcome, because the natural course of type 2 RA involves little structural damage and a lack of improvement can also be explained by the inability to measure this [5]. The long-term outcomes studied here, on the other hand, had the potential for indicating improvement, also in patients with type 2 RA.

Mortality was studied without adjusting for mortality in the general population because excess mortality in RA is heavily dependent on follow-up duration, which differs between the inclusion cohorts [40]. Although a significant improvement in mortality was observed in type 1 RA and not in type 2 RA, effect sizes were in the same direction. Analyses of longer follow-up in larger cohorts that also adjust for mortality in the general population are needed to determine if excess mortality is reduced differently between the 2 groups.

In current treatment strategies SDFR is not targeted. Although innovative, this is an interesting outcome from an immunological perspective that resembles "cure." Prolonged follow-up duration is required to determine the sustainability of DMARD-free remission after DMARD cessation. An advantage of our data is that we had median 5.5 years of follow-up after DMARD stop.

RA was defined according to the 1987 criteria (not the 2010 criteria) to exclude influences of temporal changes in rheumatologists' views on diagnosing RA. Furthermore, autoantibodies load heavily in the 2010 criteria. It is known that much inflammation is needed in the absence of autoantibodies to fulfill the 2010 criteria [16,17]. Further, in our data, higher tender joint counts were needed to classify RA in recent periods, possibly resulting in incomparability in DAS28-ESR within the current set of autoantibody-negative 2010-criteria RA patients. Nonetheless, similar results in long-term outcomes were found.

**Future implications.**   Possible implications of formal subdivision of RA are execution of more focused pathogenetic studies, development of treatment protocols adapted to disease type, and performance of trials by disease type. Ultimately a better distinction leads to improved personalized care.

## Conclusion

In sum, to our knowledge this is the first long-term study in a large cohort of RA patients with data on 25 years of follow-up. Based on the demonstrated differences in long-term outcomes,

and supported by previous findings on risk factors, we propose to subgroup RA into type 1 and type 2, in the hope that this leads to stratified treatment in RA.

## Supporting information

**S1 Checklist. STROBE Statement—checklist of items that should be included in reports of cohort studies.**
(DOCX)

**S1 Fig. Flowchart of patient inclusion.**
(DOCX)

**S2 Fig. Directed acyclic graph of causal mechanisms to identify potential sources of confounding.**
(DOCX)

**S3 Fig. Percentage of patients achieving DAS28-ESR remission (<2.6) after 1 and 3 years of follow-up in type 1 (autoantibody-positive) RA and type 2 (autoantibody-negative) RA.**
(A) Type 1 RA; (B) type 2 RA.
(DOCX)

**S4 Fig. Flowchart of RA patients fulfilling the 1987 and/or 2010 criteria.**
(DOCX)

**S5 Fig. Disease activity compared between type 1 (autoantibody-positive) and type 2 (autoantibody-negative) RA.**
(DOCX)

**S6 Fig. Sustained DMARD-free remission compared between type 1 (autoantibody-positive) and type 2 (autoantibody-negative) RA.**
(DOCX)

**S7 Fig. Mortality compared between type 1 (autoantibody-positive) and type 2 (autoantibody-negative) RA.**
(DOCX)

**S8 Fig. Functional disability compared between type 1 (autoantibody-positive) and type 2 (autoantibody-negative) RA.**
(DOCX)

**S9 Fig. Autoantibody status over time in RA patients who were autoantibody-negative at diagnosis, showing that conversion to autoantibody positivity is rare.**
(DOCX)

**S1 Table. Biologic use (prevalence within different follow-up durations) by inclusion period, showing slightly more biologic use in type 1 (autoantibody-positive) RA than type 2 (autoantibody-negative) RA.**
(DOCX)

**S2 Table. Characteristics of patients with type 1 (autoantibody-positive) and type 2 (autoantibody-negative) RA at first presentation to the EAC that fulfilled the 2010 criteria.** (A) Type 1 RA; (B) type 2 RA.
(DOCX)

**S3 Table. Disease activity during the first year and subsequent follow-up and long-term outcomes—sustained DMARD-free remission, mortality, and functional disability—by**

inclusion period compared to the reference period for type 1 (autoantibody-positive) RA fulfilling 2010 criteria.
(DOCX)

**S4 Table. Disease activity during the first year and subsequent follow-up and long-term outcomes—sustained DMARD-free remission, mortality, and functional disability—by inclusion period compared to the reference period for type 2 (autoantibody-negative) RA fulfilling 2010 criteria.**
(DOCX)

**S5 Table. Disease activity during the first year and subsequent follow-up and long-term outcomes—sustained DMARD-free remission, mortality, and functional disability—by inclusion period compared to the reference period for type 1 (autoantibody-positive) patients aged <65 years at diagnosis.**
(DOCX)

**S6 Table. Disease activity during the first year and subsequent follow-up and long-term outcomes—sustained DMARD-free remission, mortality, and functional disability—by inclusion period compared to the reference period for type 2 (autoantibody-negative) patients aged <65 years at diagnosis.**
(DOCX)

**S7 Table. Long-term outcomes in type 1 (autoantibody-positive) and type 2 (autoantibody-negative) RA after correction for left truncation.**
(DOCX)

**S1 Text. Confounding.**
(DOCX)

**S2 Text. Additional R-packages used.**
(DOCX)

## Author Contributions

**Conceptualization:** Xanthe M. E. Matthijssen, Ellis Niemantsverdriet, Tom W. J. Huizinga, Annette H. M. van der Helm–van Mil.

**Data curation:** Xanthe M. E. Matthijssen, Ellis Niemantsverdriet.

**Formal analysis:** Xanthe M. E. Matthijssen.

**Funding acquisition:** Ellis Niemantsverdriet, Annette H. M. van der Helm–van Mil.

**Investigation:** Xanthe M. E. Matthijssen, Annette H. M. van der Helm–van Mil.

**Methodology:** Xanthe M. E. Matthijssen, Ellis Niemantsverdriet, Annette H. M. van der Helm–van Mil.

**Project administration:** Xanthe M. E. Matthijssen, Ellis Niemantsverdriet, Annette H. M. van der Helm–van Mil.

**Resources:** Tom W. J. Huizinga.

**Supervision:** Tom W. J. Huizinga, Annette H. M. van der Helm–van Mil.

**Visualization:** Xanthe M. E. Matthijssen.

**Writing – original draft:** Xanthe M. E. Matthijssen, Annette H. M. van der Helm–van Mil.

**Writing – review & editing:** Ellis Niemantsverdriet, Tom W. J. Huizinga.

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
