## [Editor Report · Decision Letter 0]

3 Feb 2020

Dear Dr Matthijssen, 

Thank you for submitting your manuscript entitled "Long-term outcomes of a 25-year longitudinal cohort study indicate that Rheumatoid Arthritis can be divided in type 1 and type 2" for consideration by PLOS Medicine.

Your manuscript has now been evaluated by the PLOS Medicine editorial staff and I am writing to let you know that we would like to send your submission out for external peer review.

Kind regards,

Helen Howard, for Clare Stone PhD 

Acting Editor-in-Chief

PLOS Medicine 

plosmedicine.org

---

## [Decision Letter · Decision Letter 1]

2 May 2020

Dear Dr. Matthijssen,

Thank you very much for submitting your manuscript "Long-term outcomes of a 25-year longitudinal cohort study indicate that Rheumatoid Arthritis can be divided in type 1 and type 2" (PMEDICINE-D-20-00279R1) for consideration at PLOS Medicine. 

[LINK]

In light of these reviews, I am afraid that we will not be able to accept the manuscript for publication in the journal in its current form, but we would like to consider a revised version that addresses the reviewers' and editors' comments. Obviously we cannot make any decision about publication until we have seen the revised manuscript and your response, and we plan to seek re-review by one or more of the reviewers. 

We expect to receive your revised manuscript by May 25 2020 11:59PM. Please email us (plosmedicine@plos.org) if you have any questions or concerns.

We look forward to receiving your revised manuscript. 

Sincerely,

Emma Veitch, PhD

PLOS Medicine

On behalf of Clare Stone, PhD, Acting Chief Editor,

PLOS Medicine

plosmedicine.org

*In responding to the reviewers' points, the issue raised by the statistical reviewer (that the analysis currently does not include a between-group comparison but only within-groups) should be fully dealt with, as this seems crucial. 

*Please revise your title according to PLOS Medicine's style. Your title must be nondeclarative (ie not state the findings), rather beginning with the main concept if possible. Please place the study design in the subtitle (ie, after a colon) - eg in this case "prospective cohort". 

*In the last sentence of the Abstract Methods and Findings section, please describe the main limitation(s) of the study's methodology.

*At this stage, we ask that you include a short, non-technical Author Summary of your research to make findings accessible to a wide audience that includes both scientists and non-scientists. The Author Summary should immediately follow the Abstract in your revised manuscript. This text is subject to editorial change and should be distinct from the scientific abstract. Please see our author guidelines for more information: https://journals.plos.org/plosmedicine/s/revising-your-manuscript#loc-author-summary

*In the methods section, where you currently state that local ethical approval was obtained, would suggest also naming the ethics board/committee that gave approval (this is stated in some of the additional information with the paper, but good if it can be updated in the paper itself as well).

*Please present and organize the Discussion as follows: a short, clear summary of the article's findings; what the study adds to existing research and where and why the results may differ from previous research; strengths and limitations of the study; implications and next steps for research, clinical practice, and/or public policy; one-paragraph conclusion.

*When you cover limitations in the Discussion, it would be good to say something about the intended purpose of the instrumental variable analysis in this study. How well does the use of time period fulfill the assumptions of such an analysis? How well does this approach begin to deal with some of the limitations of a non-instrumented-for analysis?

*Please ensure that the study is reported according to the STROBE guideline, and include the completed STROBE checklist as Supporting Information. 1 Please add the following statement, or similar, to the Methods: "This study is reported as per the Strengthening the Reporting of Observational Studies in Epidemiology (STROBE) guideline (SChecklist)." The STROBE guideline can be found here: http://www.equator-network.org/reporting-guidelines/strobe/. When completing the checklist, please use section and paragraph numbers, rather than page numbers.

*Did your study have a prospective protocol or analysis plan? Please state this (either way) early in the Methods section.

Comments from the reviewers:

Reviewer #1: Alex McConnachie, Statistical Review

Matthijssen et al present an analysis of data from a large cohort of RA patients. They use period of entry to the cohort as a proxy for the type of treatment received, and analyse four long term outcomes (disease activity, remission, survival, and disability). This review considers the statistical aspects of the paper.

The use of time period as an instrumental variable is acceptable, even though the authors subsume earlier diagnosis within improved treatment protocols. The methods of analysis used for each outcome are also good.

For me, the main problem with the paper is the lack of comparison between the two disease subtypes. Results are generally presented for each outcome, within each subtype, but do not compare the two types directly. The authors conclude, based on statistically significant results within one group but not the other, that the two groups are different. This is not enough; the results should be presented in such a way as to test whether the two groups are different. There is no statistical evidence presented that supports the main conclusion of the paper.

In the abstract, it is not clear why two HRs are presented for SDFR rates in type 1 RA. Only when looking at the tables is it obvious what these HRs represent. Also, when reporting HAQ improvements, it is not clear what is being reported. Also, the confidence limits are the wrong way round (negative changes are being reported as improvements, by removing the minus signs).

Reviewer #2: Interesting but on review of this revision I think the designation of Type 1 and Type 2 is not quite right. Really this is just ACPA and/or RF positive vs. ACPA and/or RF negative RA and without trying to bring in other autoantibodies, even if not standard in RA diagnostic yet (e.g. anti-CarP, ANAs, other), or features of disease (nodules etc) it seems too much of a stretch to label these subsets Type 1 and 2 and trying to make a case for any nomenclature change. I think still very interesting but perhaps revise to just state that this is analyses of ACPA and/or RF positive vs. negative, and then in the discussion section discuss how this may for the basis for future classification. 

Literature supports that subjects who are seroneg at initial diagnosis may later become positive. was this evaluated?

please list the specific ACPA and RF assays used and also comment on how seropositivity may differ based on the test used

Please provide analyses of how various elements of disease activity measures may differ between disease subsets. for example, is CRP or swollen joints comprising more of the score in one group vs. global scores in another

please discuss how diagnostic certainty may influence the use of biologics - this can be separate issue from disease severity. for example, a practitioner may be more willing to use expensive and perhaps toxic meds for seropositive RA because they are sure of their diagnosis based on the blood test.

1987 criteria don't include ACPA - how was this handled in RF negative, ACPA positive subjects? could they have been excluded because they may not have met criteria?

Reviewer #3: Matthijssen and co-authors report here on the long term outcomes of autoantibody-positive and -negative rheumatoid arthritis (RA) over 15 years of follow-up. Outcomes of interest are drug-free remission, functional disability and mortality. Data were retrieved from the large Leiden Early Arthritis inception cohort. All analyses were adjusted for the inclusion period, taking into account important modifications that have occurred in the management of RA over the years (earlier diagnosis, use of methotrexate as first DMARD, treat-to-target strategies). Collectively, the Authors show that all the outcomes analyzed ameliorated over time in autoantibody-positive but not autoantibody-negative RA.

Data arising from the current research work are of interest as they open the interesting perspective on whether autoantibody-positive and -negative RA should be managed as two different diseases entities. However, a number of confounders should be carefully evaluated before drawing definitive conclusions.

Major points to consider:

1) All the outcomes analyzed are strongly dependent of the treatment strategy adopted. Although the Authors show similar trends in the reduction of disease activity in autoantibody-positive and -negative patients, it cannot be excluded that differences in the observed outcomes may depend on different treatment strategies. Patients with autoantibody-negative RA may indeed receive less frequent combination therapy with biological drugs. This appears clearly evident from Table S1. Also the use of glucocorticoids may differ. It is therefore important that the Authors provide information on (and correct their analyses for):

- the trends in drug-free remission, functional disability and mortality in patients treated with csDMARDs and in those also receiving bDMARDs stratified for the autoantibody status. Indeed, I believe that adjusting analyses for inclusion period may not take into account the use of bDMARDs. From Table S1, it appears that bDMARDs have increased in autoantibody-positive but not in -negative RA in more recent times.

- the use of glucocorticoids in autoantibody-positive and -negative patients

2) As the Authors acknowledge, autoantibody-negative RA may itself be a heterogeneous disease entity. Among the various confounders, elderly-onset RA has peculiar clinical features in presentation and outcomes. From Table 1, it appears autoantibody-negative patients are (as expected) older. I would therefore suggest that the Authors perform a subanalysis on patients (both autoantibody-positive and -negative) aged less than 65 years. I am aware that this could limit the frequency of observed events for the outcome of mortality, but other outcomes remain valid.

Additional points:

The scope of the current study was a within-subgroup comparison of outcomes over time, and not a direct comparison between autoantibody-positive and -negative RA. However, from the Kaplain Maier curves presented, it seems that mortality and functional disability have remained poorer in autoantibody-negative RA compared to autoantibody-positive even after the adoption of T2T. To avoid misinterpretations, it would be important to provide the reader with numbers of in the Kaplan Meier curves. Is the percentage of losts to follow-up different between autoantibody-positive and -negative RA? This would help understanding whether the current analyses have selected particular patient populations.

[LINK]

---

## [Decision Letter · Decision Letter 2]

6 Jul 2020

Dear Dr. Matthijssen,

Thank you very much for re-submitting your manuscript "Differences in improvement of disease outcomes between autoantibody-positive and autoantibody-negative rheumatoid arthritis with enhanced treatment strategies over 25 years: A longitudinal prospective cohort study" (PMEDICINE-D-20-00279R2) for review by PLOS Medicine.

I have discussed the paper with my colleagues and the academic editor and it was also seen again by two reviewers. I am pleased to say that provided the remaining editorial and production issues are dealt with we are planning to accept the paper for publication in the journal.

[LINK]

We look forward to receiving the revised manuscript by Jul 13 2020 11:59PM. 

Sincerely,

Artur Arikainen, 

Associate Editor 

PLOS Medicine

plosmedicine.org

Requests from Editors:

1. Please update your title to: "Enhanced treatment strategies and distinct disease outcomes among autoantibody-positive and -negative rheumatoid arthritis patients over 25 years: a longitudinal cohort study in the Netherlands"

2. Please add line numbers in the margin throughout your manuscript.

3. The Data Availability Statement (DAS) requires revision. For each data source used in your study: 

4. Abstract:

a. Please quantify your results with 95% CIs and p values.

b. Please provide a p value here or in the main Results section to support this statement: “…percentage of autoantibody-positive patients were constant throughout the inclusion periods”, as it appears to have varied over time.

c. Please remove or replace “idem”.

d. Please include another limitation at the end of ‘Methods and findings’, eg. limited spread of ethnicities.

e. Please consider add the following to the Conclusion (from your main Conclusion section): “…, in the hope that this leads to stratified treatment in RA.”

5. Author Summary:

a. Add “(RA)” after the first instance of ‘rheumatoid arthritis’.

b. Please clarify ‘histology’ for a non-scientist audience.

c. Please correct typo: “distict”

6. Please delete this sentence, as it is somewhat irrelevant to RA: “Careful clinical observations over time have led to the description of diseases, originally by the name of the doctor who made the observation.”

7. Please move the reference callouts to before punctuation, and do not include spaces within the callout, eg: “…different environmental risk factors [3,4],…”

8. Methods:

a. Please include the exact date ranges (including day and month) for each period of patient recruitment/inclusion.

b. Please replace “As suggested by the reviewer,…” with “In response to requests during peer review,…”

c. Please mention explicitly that there was no prospective study protocol or analysis plan.

9. Results:

a. Please quantify your results with 95% CIs and p values.

b. In the Figures and Tables, please label the groups as “autoantibody-positive” and “autoantibody-negative” (or similar), rather than (or in addition to) types 1 and 2, since your study appears to be introducing this typology.

10. Given that your study design cannot show causation, please replace all instances of “effect” with “association” or similar.

11. The terms gender and sex are not interchangeable (as discussed in http://www.who.int/gender/whatisgender/en/ ); please use the appropriate term.

12. Discussion:

a. Please make the first paragraph a summary of your study’s findings.

b. Please correct typo: “Advantageous”

13. PLOS does not permit references to unpublished data. Please remove any such claims, or do one of the following:

a) If you are the owner of the data relevant to this claim, please provide the data in accordance with the PLOS data policy, and update your Data Availability Statement as needed.

b) If the data not shown refer to a study from another group that has not been published, please cite personal communication in your manuscript text (it should not be included in the reference section). Please provide the name of the individual, the affiliation, and date of communication. The individual must provide PLOS Medicine written permission to be named for this purpose.

c) For any other circumstance, please contact the journal office ASAP.

14. Please remove the Footnotes section – all relevant information should be included on the submission form or in the main manuscript text instead.

15. Table 1 footnotes contain an extra semicolon.

16. Table 1: Please define EAC.

17. Figure 1 (and check others): Ensure all abbreviations are defined, eg. MTX, T2T.

18. Tables 2, 3, 4: Please give exact p values where p>0.001.

19. Please upload the STROBE checklist as a separate file, named “S1 Checklist”.

----------

Comments from Reviewers:

Reviewer #1: Alex McConnachie, Statistical Review

I thank the authors for their responses to my original comments. Comparing the changes over time in outcomes between the two groups of patients is very useful. However, the authors now need to use these findings in their conclusions.

Table 4 shows quite clearly that the rate of decline of DAS28 after the first year is greater in Type 1 patients.

However, there is no evidence to suggest any difference between groups in the rate at which mortality has reduced over time. This needs to be carried forward into the conclusions of the paper. Many of the results are reported and interpreted as outcomes improving in Type 1 but not Type2 patients, but the results shown in Table 4 show no evidence of a difference between the groups for mortality. Looking at Tables 2 and 3, the hazard ratios relative to the earliest time period are all less than 1, and in some instances lower for Type 2 than for Type 1 patients, but the differences between them are not large. The fact that in one group, the hazard ratios are significantly lower than one, whilst in the other they are not, may simply reflect the different sample sizes. In order to claim a difference between the groups, the statistical evidence (as shown in Table 4) should be referred to.

It is interesting, for all the other outcomes, that there appear to be significantly greater improvements in the 2006-10 period for Type 1 compared to Type 2 patients (even for DAS28 improvement during the first year of treatment, the difference is on the borderline of p=0.05), and whilst none of the interactions at other time points reach formal statistical significance, there appear to be clear trends towards better improvements in all outcomes other than mortality from 2001 to 2016. However, other than for the rate of change in DAS28 after the first year of treatment, there is little to suggest any differences between the groups in response to treatment in 1997-2000.

As stated above, when discussing the differences between groups in the improvements over time, the authors cannot simply report a significant improvement for one group and not the other, and draw conclusions from that. They must refer to the results in Table 4 to determine for which outcomes, and during which time periods, there is evidence of a difference between the two groups.

Finally, it may be useful to report p-values from likelihood ratio tests, comparing models with and without interaction terms, to give a single p-value for each outcome, to assess whether the changes over time differ between the two groups. This would be in addition to the information already reported in Table 4; maybe a separate row.

Reviewer #3: The authors have addressed my major points. Non further revision is required

[LINK]

---

## [Editor Report · Decision Letter 3]

18 Aug 2020

Dear Dr. Matthijssen, 

On behalf of my colleagues and the academic editor, Dr. Carlomaurizio Montecucco, I am delighted to inform you that your manuscript entitled "Enhanced treatment strategies and distinct disease outcomes among autoantibody-positive and -negative rheumatoid arthritis patients over 25 years: a longitudinal cohort study in the Netherlands" (PMEDICINE-D-20-00279R3) has been accepted for publication in PLOS Medicine. 

PRODUCTION PROCESS

PRESS

PROFILE INFORMATION

Thank you again for submitting the manuscript to PLOS Medicine. We look forward to publishing it. 

Best wishes, 

Artur Arikainen, 

Associate Editor 

PLOS Medicine

plosmedicine.org